# The off-hour effect on mortality in traumatic brain injury according to age group

**Eujene Jung[1], Hyun Ho Ryu[2]***

1 Department of Emergency Medicine, Chonnam National University Hospital, Gwangju, South Korea,
2 Chonnam National University College of Medicine, Gwangju, South Korea

* em.ryu.hyunho@gmail.com

**Data Availability Statement:** The data for this study cannot be made freely available online as they are patient-related and public availability would compromise study participants' privacy. Data are available upon request from CNUHEM

## Abstract

### Background

Traumatic brain injury (TBI) is a time-sensitive and life-threatening medical condition. We hypothesized that off-hours, which includes night-time, weekends, and holidays, may influence mortality in TBI. Our study aimed to evaluate if the off-hours effect influences mortality in patients with TBI and whether this effect is dependent on the age group.

### Methods

This study included patients who experienced TBI and were admitted to Chonnam National University Hospital (CNUH) between 2017 to 2020. The main exposure was arrival time at the emergency department (ED) (off-hours vs. working hours). The main outcome was mortality at hospital discharge. Multivariable logistic regression analysis was conducted to estimate the effect size of off-hours on mortality compared to that of working hours. We performed an interaction analysis between ED admission time and age group on study outcomes.

### Results

A total of 2086 patients with TBI with intracranial injury who were transported by EMS were enrolled in our registry. In the multivariable logistic regression analysis, there was no significant difference in mortality (AOR, 95% CI (1.05 [0.54–1.81]) in patients visiting the ED during off-hours. In the interaction analysis, the effect measure of ED admission during off-hours on mortality was significant among younger people (0–17 years: 1.16 [1.03–1.31]), compared to that in other age groups (18–64 years: 1.02 [0.48–2.39] and 65–100 years (0.99 [0.51–2.23])).

### Conclusions

In patients under 18 years old, admission during off-hours was associated with higher mortality at hospital discharge compared to admission during working-hours in patients with TBI with intracranial hemorrhage. EDs should be designed such that the same quality of emergency care is provided regardless of admission time.

data center by phone (82-622206809) or email
(cnuhem@gmail.com) for researchers who meet
the criteria for access to confidential data.

**Funding:** This study was supported by a grant
(BCRI-20022) of Chonnam National University
Hospital Biomedical Research Institute.The funders
had no role in study design, data collection and
analysis, decision to publish, or preparation of the
manuscript.

**Competing interests:** The Authors have declared
that no competing interests exists.

## Introduction

Traumatic brain injury (TBI) is an important public health concern. Approximately 2.8 million individuals sustain TBI annually, approximately 280,000 are hospitalized, and 50,000 TBI-related mortalities are reported, which is one-third of all injury-related deaths in the US [1].

TBI is a time-sensitive and life-threatening medical condition [2,3]. If left untreated, TBI can lead to fatal complications resulting from increased intracranial pressure and brain herniation [4,5]. Surgical and non-surgical approaches to TBI management aim to reduce symptoms and improve outcomes by relieving intracranial hemorrhage [4,6]. Previous studies have reported that rapid surgical intervention with four hours after injury markedly lowered the mortality rate and favorable functional recovery [7]. Thus, surgical and non-surgical interventions for the treatment of TBI are the most effective in improving outcomes if performed promptly.

Differences in the quality of care during various working hours in hospitals, such as during weekdays and on weekends and during regular working hours and off-hours, have been studied in the context of various diseases and settings [8,9]. During off-hours, most hospitals have a tendency to reduce employee levels [10]. Additionally, due to less hospital staffing, particularly the more experienced doctor and certain investigational services may not be immediately available during off-hours and weekends [11]. A study in the US showed that mortality rates in patients with trauma admitted at night were 1.18 times that of those presenting within working hours [12] However, some studies found the weekend effect to be controversial in patients with trauma [13,14]; in a single-center study of Japan, the mortality in patients with trauma requiring subspecialty intervention was not different between those admitted during working hours and those admitted during off-hours. The impact on the outcomes of the patients with trauma transported urgently was, however, not evaluated sufficiently according to time of day and day of the week [15]. In a study among old patients with TBI in the US, those aged between 65 and 89 who visited hospitals on weekends had a 13% higher risk of mortality [16].

There have been no conclusive reports as to whether the off-hour impact on TBI is a truth or a myth. Therefore, we hypothesized that the off-hour effect may influence mortality in TBI, and this effect may depend on demographics, such as the age of the patients. Our study aimed to evaluate whether the off-hour effect influences mortality in patients with TBI and whether this effect is dependent on patient age group.

## Methods

### Study design, setting, and population

This retrospective study included patients who experienced TBI and were admitted to Chonnam National University Hospital (CNUH) between January 2017 and December 2020.

CNUH trauma center covers Gwang-ju and Chonnam provinces (population: 3.35 million inhabitants per 12,678 km$^2$). The central Fire Services (CFS) provides fire-based and public services under 25 Emergency Medical Service (EMS) agencies with 144 ambulances and a single unified dispatch center in each province, with approximately 200,000 transports per year.

Emergency medical technicians (EMTs) in Korea are classified into level-1 and level-2 EMTs (comparable to EMT-intermediate and EMT-basic in the US, respectively). The level-1 EMTs can be performed with high grade procedures including advanced airway and intravenous line access compared to level-2 EMTs. EMTs can perform vital sign check, neck brace, oxygen administration, and fluid resuscitation under direct medical control at the scene and during transport. When severe trauma is suspected, the patient is transferred to the level 1 trauma center under the judgement of EMSs and physicians. Most of the EMS teams are made

up of three paramedics, and the same number of paramedics are present at night and during weekends and holidays.

The CNUH trauma center was established in 2010 and has since employed a team of expert physicians in various departments including neurosurgery, general surgery, thoracic surgery, orthopedic surgery, radiology, and emergency medicine. If a patient is suspected to have major trauma, including TBI, by the EMS paramedics, it is reported to the hospital by paramedics at the trauma scene or during transport. After arriving at the hospital, physical examinations, including neurologic examination, laboratory tests, and brain/abdomen/chest computed tomography (CT), are executed as soon as possible. Emergency surgery is performed (all year round) if an injury requiring surgical treatment is diagnosed in the radiologic examination. After surgery, depending on the patient's condition, the patient is hospitalized in a specialized trauma ward or trauma intensive care unit. These protocols are the same regardless of age, and there is no medical staff dedicated to a specific age group.

The present study was approved by the Institutional Review Board of CNUH University Hospital. Given the retrospective nature of the study and the use of anonymized patient data, the requirement for informed consent were waived.

The inclusion criteria are patients with TBI who visited trauma center using emergency medical services (EMS) within 72 hours of injury and had intracranial hemorrhage and/or diffuse axonal injury confirmed by a radiological examination. Patients who had TBI accompanied by injuries at other sites, those with no information of hospital admission, those who underwent surgical intervention, and those who died were excluded. Patients with neurological disorders, psychiatric disorders, and terminal cancer, pregnant women, patients with penetrating brain injury, and patients transferred to the participating hospital ED after surgery at another hospital were also excluded from the study.

## Main outcome and variables

The main outcome measure was mortality at hospital discharge. The secondary outcome was surgical intervention within 6 hours of ED admission. The main exposure variable was arrival time at ED admission (off-hour vs. working-hour). We defined daytime as 09:00 AM to 17:59 PM and nighttime as 18:00 PM to 08:59 AM; weekdays as Monday to Friday except national holidays; and weekends as Saturday, Sunday, and national holidays, based on a previous study [17]. We defined off-hour when it was nighttime or weekends, and defined other times as working-hour.

We included data regarding patient's demographics (age, sex, and comorbidities [hypertension and diabetes mellitus]), intentionality (including self-inflicted and interpersonal acts of violence intended to cause injury), injury characteristics (mechanism of injury, place of injury, and injury severity score), clinical findings (initial vital signs at ED admission, type of intracranial hemorrhage found on brain imaging, treatment in the ED, surgical or non-surgical treatment, and disposition after ED treatment), and patient outcomes at the time of hospital discharge.

## Statistical analysis

Patient demographics, prehospital and hospital variables, and study outcomes according to arrival time at ED admission were compared using the Chi-squared test for categorical variables and Wilcoxon rank-sum test for continuous variables. Multivariable logistic regression analyses were conducted to estimate the effect size of off-hours on mortality compared to that of working hours. Adjusted odds ratios (AORs) with corresponding 95% confidence intervals (CIs) were calculated after adjusting potential confounders. The potential confounders were

selected based on directed acyclic graph (DAG) models. Additionally, we conducted a subgroup analysis for patients with subdural hemorrhage. Finally, we performed interaction analysis between ED admission time and age group on study outcomes. All statistical analyses were performed using SAS version 9.4 (SAS Institute Inc., Cary, NC, USA). A two-sided significance level of 0.05 (p<0.05) was used to determine statistical significance.

## Results

### Demographic findings

During the study period, a total of 2086 patients with TBI with intracranial injury who were transported by EMS were enrolled in our registry.

The characteristics of the study population according to arrival time at the ED (time of day and day of week) are shown in Table 1. Of the study population, 40.8% (851/2,086) of the patients visited the ED during the daytime; 59.2% (1,235/2086), nighttime; 38.8% (809/2,086), weekdays; and 51.2% (1,277/2086) during weekends or holidays. The proportion of mortality was 4.7% (97/2.086) in the overall study population; 4.8% (41/851) in the daytime-visiting group; 4.5% (56/1,235) in the nighttime vising group; 4.1% (33/809) in the weekday-visiting group; and 5.0% (64/1,277) in the weekend/holiday-visiting group.

The characteristics of the study population according to arrival time at the ED (off-hours vs. working hours) are shown in Table 2. Of the study population, 84.9% (1,771/2,086) visited the ED during off-hours and 15.1% (315/2,086) patients visited during the working hours. The proportion of mortality was 4.7% (83/1,771) in the off-hours-visiting group and 4.4% (14/315) in the working-hours-visiting group. The proportion of surgical intervention was 24.0% (425/1,771) in the off-hours-visiting group and 25.7% (81/315) in the working-hours-visiting group.

### Main results

On multivariable logistic regression analysis, patient visits during off-hours showed no significant difference in mortality and surgical interventions compared to visits during working hours. The OR for mortality in patients visiting during off-hours was 1.05 (95% CI, 0.54–1.81), and it was 1.01 (0.74–1.32) for those undergoing surgical intervention (Table 3).

### Interaction analysis

There were interaction effects between arrival time at ED (working hours vs. off-hours) and age group on mortality in patients with TBI. The effect measure of ED admission during off-hours was significant in the young patients aged 0–17 years (adjusted OR [95% CI]: 1.16 [1.03–1.31]), whereas it was not significant in the 18–64 (1.02 [0.48–2.39]) and the 65–100 years age groups (0.99 [0.51–2.23]). There were no interaction effects between arrival time at ED (working hours vs. off-hours) and age group on surgical intervention.

In a subgroup analysis in patients with subdural hemorrhage (SDH), there were no interaction effects between arrival time at ED (working hours vs. off-hours) and age on mortality and surgical intervention (Table 4).

## Discussion

This study evaluated the association between arrival time at ED (off-hours vs. working hours) and mortality and surgical intervention among patients with TBI with intracranial injury and found that the off-hour effect was not observed in patients admitted to the level 1 trauma center. However, in the younger group aged <18 years, mortality was significantly higher during

**Table 1. Characteristics of the study population according to the arrival time (Time of day and Day of week) of emergency department.**

| Variables | All | Arrival time of Emergency department | | | | | |
|---|---|---|---|---|---|---|---|
| | | Time of day | | | Day of week | | |
| | N (%) | Daytime | Night time | p-value | Weekdays | Weekends/holidays | p-value |
| All | 2,086 (100.0) | 851 (100.0) | 1,235 (100.0) | | 809 (100.0) | 1,277 (100.0) | |
| Age | | | | <0.01 | | | 0.03 |
| 0–17 | 204 (9.8) | 83 (9.8) | 121 (9.8) | | 81 (10.0) | 123 (9.6) | |
| 18–64 | 1,207 (57.9) | 401 (47.1) | 806 (65.3) | | 495 (61.2) | 712 (55.8) | |
| 65- | 675 (32.4) | 367 (43.1) | 308 (24.9) | | 233 (28.8) | 442 (34.6) | |
| Sex, female | 708 (33.9) | 320 (37.6) | 388 (31.4) | <0.01 | 266 (32.9) | 442 (34.6) | 0.65 |
| Hypertension | 642 (30.8) | 255 (30.0) | 387 (31.3) | 0.17 | 259 (32.0) | 383 (30.0) | 0.05 |
| Diabetes mellitus | 493 (23.6) | 195 (22.9) | 298 (24.1) | 0.44 | 202 (25.0) | 291 (22.8) | 0.11 |
| Intentionality, yes | 332 (15.9) | 75 (8.8) | 257 (20.8) | <0.01 | 142 (17.6) | 190 (14.9) | 0.02 |
| Mechanism of injury | | | | <0.01 | | | 0.15 |
| Traffic | 746 (35.8) | 302 (35.5) | 444 (36.0) | | 291 (36.0) | 455 (35.6) | |
| Fall down | 820 (39.3) | 389 (45.7) | 431 (34.9) | | 321 (39.7) | 499 (39.1) | |
| Stuck and hit by person or object | 275 (13.2) | 68 (8.0) | 207 (16.8) | | 114 (14.1) | 161 (12.6) | |
| Other | 245 (11.7) | 92 (10.8) | 153 (12.4) | | 83 (10.3) | 162 (12.7) | |
| Place of injury | | | | <0.01 | | | <0.01 |
| Home | 587 (28.1) | 260 (30.6) | 327 (26.5) | | 212 (26.2) | 375 (29.4) | |
| Street | 975 (46.7) | 366 (43.0) | 609 (49.3) | | 397 (49.1) | 578 (45.3) | |
| Industrial/construction | 84 (4.0) | 53 (6.2) | 31 (2.5) | | 19 (2.3) | 65 (5.1) | |
| Commercial | 192 (9.2) | 57 (6.7) | 135 (10.9) | | 80 (9.9) | 112 (8.8) | |
| Other | 248 (11.9) | 115 (13.5) | 133 (10.8) | | 101 (12.5) | 147 (11.5) | |
| Alcohol drinking before injury, yes | 463 (22.2) | 50 (5.9) | 413 (33.4) | <0.01 | 217 (26.8) | 246 (19.3) | <0.01 |
| Advanced airway, yes | 110 (5.3) | 47 (5.5) | 63 (5.1) | 0.67 | 38 (4.7) | 72 (5.6) | 0.4 |
| Hypotension (<90mmHg), yes | 88 (4.2) | 33 (3.9) | 55 (4.5) | <0.52 | 40 (4.9) | 48 (3.8) | 0.49 |
| Heart rate | | | | <0.01 | | | 0.13 |
| Bradycardia (<60 beats per min) | 73 (3.5) | 33 (3.9) | 40 (3.2) | | 38 (4.7) | 35 (2.7) | |
| Tachycardia (>100 beats per min) | 532 (25.5) | 178 (20.9) | 354 (28.7) | | 199 (24.6) | 333 (26.1) | |
| Mentality | | | | 0.53 | | | <0.01 |
| Alert | 1,822 (87.3) | 746 (87.7) | 1,076 (87.1) | | 708 (87.5) | 1,114 (87.2) | |
| Drowsy | 64 (3.1) | 23 (2.7) | 41 (3.3) | | 32 (4.0) | 32 (2.5) | |
| Stupor | 54 (2.6) | 26 (3.1) | 28 (2.3) | | 21 (2.6) | 33 (2.6) | |
| Coma | 146 (7.0) | 56 (6.6) | 90 (7.3) | | 48 (5.9) | 98 (7.7) | |
| Severity (AIS>3) | 687 (32.9) | 330 (38.8) | 357 (28.9) | <0.01 | 254 (31.4) | 433 (33.9) | 0.18 |
| Type of hemorrhage | | | | <0.01 | | | 0.04 |
| SDH | 563 (27.0) | 333 (39.1) | 230 (18.6) | | 222 (27.4) | 341 (26.7) | |
| SAH | 522 (25.0) | 250 (29.4) | 272 (22.0) | | 212 (26.2) | 310 (24.3) | |
| ICH | 413 (19.8) | 105 (12.3) | 308 (24.9) | | 157 (19.4) | 256 (20.0) | |
| EDH | 289 (13.9) | 104 (12.2) | 185 (15.0) | | 112 (13.8) | 177 (13.9) | |
| IVH and Other | 299 (14.3) | 59 (6.9) | 240 (19.4) | | 106 (13.1) | 193 (15.1) | |
| Study outcomes | | | | | | | |
| Surgical intervention, yes | 506 (24.3) | 231 (27.1) | 275 (22.3) | 0.01 | 191 (23.6) | 315 (24.7) | 0.28 |
| Mortality at hospital discharge | 97 (4.7) | 41 (4.8) | 56 (4.5) | 0.76 | 33 (4.1) | 64 (5.0) | 0.18 |

AIS, Abbreviated Injury Score; SDH, subdural hematoma; SAH, subarachnoid hematoma; ICH, intracranial hematoma; EDH, epidural hematoma; IVH, intraventricular hematoma.

**Table 2. Characteristics of the study population according to the arrival time (Off-hours vs. Working-hours) of emergency department.**

| Variables | All | Arrival time of emergency department | | |
|---|---|---|---|---|
| | N (%) | Off-hours | Working-hours | p-value |
| All | 2,086(100.0) | 1,771(100.0) | 315(100.0) | |
| Age | | | | <0.01 |
| 0–17 | 204(9.8) | 166(9.4) | 38(12.1) | |
| 18–64 | 1,207(57.9) | 1,050(59.3) | 157(49.8) | |
| 65- | 675(32.4) | 555(31.3) | 120(38.1) | |
| Sex, female | 708(33.9) | 597(33.7) | 111(35.2) | 0.6 |
| Hypertension | 642 (30.8) | 620 (35.0) | 22 (7.0) | <0.01 |
| Diabetes mellitus | 493 (23.6) | 442 (25.0) | 51 (16.2) | <0.01 |
| Intentionality, yes | 332(15.9) | 308(17.4) | 24(7.6) | <0.01 |
| Mechanism of injury | | | | <0.01 |
| Traffic | 746(35.8) | 630(35.6) | 116(36.8) | |
| Fall down | 820(39.3) | 676(38.2) | 144(45.7) | |
| Stuck and hit by person or object | 275(13.2) | 247(13.9) | 28(8.9) | |
| Other | 245(11.7) | 218(12.3) | 27(8.6) | |
| Place of injury | | | | 0.48 |
| Home | 587(28.1) | 501(28.3) | 86(27.3) | |
| Street | 975(46.7) | 828(46.8) | 147(46.7) | |
| Industrial/construction | 84(4.0) | 70(4.0) | 14(4.4) | |
| Commercial | 192(9.2) | 169(9.5) | 23(7.3) | |
| Other | 248(11.9) | 203(11.5) | 45(14.3) | |
| Alcohol drinking before injury, yes | 463(22.2) | 446(25.2) | 17(5.4) | <0.01 |
| Advanced airway, yes | 110(5.3) | 92(5.2) | 18(5.7) | 0.7 |
| Hypotension (<90mmHg), yes | 88(4.2) | 73(4.1) | 15(4.8) | 0.6 |
| Heart rate | | | | 0.08 |
| Bradycardia (<60 beats per min) | 73(3.5) | 57(3.2) | 16(5.1) | |
| Tachycardia (>100 beats per min) | 532(25.5) | 464(26.2) | 68(21.6) | |
| Mentality | | | | 0.63 |
| Alert | 1,822(87.3) | 1,544(87.2) | 278(88.3) | |
| Drowsy | 64(3.1) | 53(3.0) | 11(3.5) | |
| Stupor | 54(2.6) | 45(2.5) | 9(2.9) | |
| Coma | 146(7.0) | 129(7.3) | 17(5.4) | |
| Severity (AIS>3) | 687(32.9) | 575(32.5) | 112(35.6) | 0.28 |
| Type of hemorrhage | | | | 0.04 |
| SDH | 563 (27.0) | 480(27.1) | 83(26.3) | |
| SAH | 522 (25.0) | 454 (25.6) | 68 (21.6) | |
| ICH | 413 (19.8) | 338 (19.1) | 75 (23.8) | |
| EDH | 289 (13.9) | 247 (13.9) | 42 (13.3) | |
| IVH and Other | 299 (14.3) | 252 (14.2) | 47 (14.9) | |
| Study outcomes | | | | |
| Surgical intervention, yes | 506(24.3) | 425(24.0) | 81(25.7) | 0.51 |
| Mortality at hospital discharge | 97(4.7) | 83(4.7) | 14(4.4) | 0.85 |

AIS, Abbreviated Injury Score; SDH, subdural hematoma; SAH, subarachnoid hematoma; ICH, intracranial hematoma; EDH, epidural hematoma; IVH, intraventricular hematoma.

**Table 3. Multivariable logistic regression analysis on study outcomes by the arrival time at emergency department.**

| | Total | Outcome | | Model 1 | Model 2 |
|---|---|---|---|---|---|
| | N | N | % | AOR (95% CI) | AOR (95% CI) |
| Mortality at hospital discharge | | | | | |
| Arrival time at emergency department | 2086 | 97 | 4.7 | | |
| Working-hours | 315 | 14 | 4.4 | Ref. | Ref. |
| Off-hours | 1771 | 83 | 4.7 | 1.04 (0.57–1.94) | 1.05 (0.54–1.81) |
| Age | | | | | |
| 0–17 | 204 | 3 | 1.5 | 0.31 (0.14–1.10) | 0.41 (0.14–1.31) |
| 18–64 | 1207 | 52 | 4.3 | Ref. | Ref. |
| 65- | 675 | 42 | 6.2 | 1.49 (0.94–2.25) | 1.77 (1.12–2.74) |
| Surgical intervention | | | | | |
| Arrival time at emergency department | 2086 | 506 | 24.3 | | |
| Working-hours | 315 | 81 | 25.7 | Ref. | Ref. |
| Off-hours | 1771 | 425 | 24.0 | 0.92 (0.73–1.24) | 1.01 (0.74–1.32) |
| Age | | | | | |
| 0–17 | 204 | 42 | 20.6 | 0.84 (0.59–1.22) | 0.84 (0.53–1.23) |
| 18–64 | 1207 | 283 | 23.4 | Ref. | Ref. |
| 65- | 675 | 181 | 26.8 | 1.25 (0.97–1.55) | 1.19 (0.92–1.51) |

Ref., reference

Model 1: Adjusted for age, sex, hypertension, and diabetes mellitus.

Model 2: Adjusted for variables of Model 1 and intentionality, mechanism of injury, and place of injury.

**Table 4. Interaction analysis between the arrival time at emergency department and age group.**

| Whole TBI patients | Arrival time | Mortality at hospital discharge | Surgical intervention |
|---|---|---|---|
| Age | | | |
| 0–17 | | | |
| | Working-hours | reference | reference |
| | Off-hours | 1.16 (1.03–1.31) | 1.54 (0.59–3.98) |
| 18–64 | | | |
| | Working-hours | reference | reference |
| | Off-hours | 1.02 (0.48–2.39) | 1.06 (0.71–1.59) |
| 65- | | | |
| | Working-hours | reference | reference |
| | Off-hours | 0.99 (0.51–2.23) | 0.89 (0.56–1.36) |
| Subdural hemorrhage patients | | | |
| Age | | | |
| 0–17 | | | |
| | Working-hours | reference | reference |
| | Off-hours | 1.15 (0.98–1.34) | 1.47 (0.57–3.81) |
| 18–64 | | | |
| | Working-hours | reference | reference |
| | Off-hours | 0.99 (0.43–2.13) | 0.91 (0.50–1.38) |
| 65- | | | |
| | Working-hours | reference | reference |
| | Off-hours | 1.04 (0.51–2.52) | 0.89 (0.56–1.36) |

off-hours (adjusted OR [95% CI]: 1.17 [1.04–1.30]). These results suggest that it is still a potential risk factor of TBI-related mortality in young patients, although the off-hour effect did not affect mortality or surgical intervention in the overall study population.

Several studies have shown that patients admitted to the ED at night or during weekends experience poorer clinical outcomes than those admitted during the daytime or weekdays [18–22]. This phenomenon is called the "weekend effect" or "off-hour effect," and it is defined as the differences in clinical outcomes experienced by patients admitted on a weekday versus those experienced by patients admitted during the weekend, or during daytime versus during nighttime. Despite the numerous studies presenting relatively poor clinical outcomes for off-hour-admitted patients, there are various possible explanations for the reason for the off-hour effect, but these are not yet clear [23].

In previous reports, the effect of off-hour ED arrival on hospital mortality was quite heterogeneous according to each disease: cervical trauma (not significant), stroke (not significant), acute kidney injury (OR: 1.07), atrial fibrillation (OR: 1.23), thoracic aortic aneurysm (OR: 2.55), abdominal aortic aneurysm (OR: 1.32), non-traumatic SDH (OR: 1.19), subarachnoid hemorrhage (SAH) (not significant), and intracerebral hemorrhage (ICH) (OR: 1.12) [24–31].

There were three possible explanations for the off-hour effect on mortality. First, human factors might be related to worse prognoses among patients with trauma during off-hours. Emergency physicians have reported negative impacts of shift work, including poor quality of sleep, irritability, fatigue, and mood decrement [32]. Second, patients admitted on weekends or at night could have been more critically ill than those admitted during the daytime and on weekdays [33,34]. Third, the off-hour effect could also be an "organization issue" due to the reduced number of staff or levels of staffing [28,35].

Contrary to the results of previous studies that analyzed the off-hour effect in several diseases, the increase in mortality was not significant in patients with TBI. However, among patients aged 0–17 years, those with TBI who arrived at the ED during off-hours showed higher odds of mortality compared to those arriving at the ED during working hours. To our knowledge, although no studies have assessed the association between pediatric TBI outcomes and time of day or day or week, our study results are consistent with that of a study that reported an off-hour effect on mortality in pediatric patients with trauma [36].

Although there are no existing studies on the reason why pediatric patients with TBI were significantly affected by the off-hour effect, it can be assumed that the lack of resources during off-hours, such as manpower, could have more seriously affected young patients because TBI pathology is age-dependent and young patients with TBI have fewer contusions, whereas severe brain injuries such as SDH and diffuse cerebral edema are more common [37].

SDH, which is reported to have a decisive effect on the prognosis of surgical treatment to decrease intracranial pressure, was assumed to be relatively disadvantage in off-hour compared to other brain injury, and a subgroup analysis was performed only on SDH patients [38–41]. In our study, although there was no statistically significant difference in the types of brain hemorrhage of patients visiting off-hours and working-hours, subgroup analysis for patients with SDH was performed, because the SDH proportion of off-hours was higher than working hours.

In the subgroup analysis of traumatic SDH, which was judged to have been largely affected by the off-hour effect due to the need for surgical treatment, no off-hour and no interaction effects with age group were observed. This is consistent with the results of previous studies that reported the weekend effect in patients with trauma with SDH, which demonstrated that patients admitted over weekends had similar mortality despite higher severity compared to patients admitted on weekdays [14].

In this study, although the number of patients aged 18–64 years was relatively increased due to the increase in alcohol drink and the increase in outdoor activities during off-hours, the off-hour effect was significantly associated with mortality at hospital discharge in the young patients under 17 year-old of age with TBI. To reduce the TBI burden, especially in young patients, additional manpower and equipment input in off-hours should be considered according to the circumstances of each hospital for maintained the quality of treatment regardless of the day and time of hospital admission.

Our study results suggest the possibility that the quality of treatment may still differ depending on arrival time at the ED in a specific patient group, although efforts are being undertaken to improve healthcare delivery to reduce the gaps in healthcare outcomes, regardless of whether patients are admitted during the daytime, nighttime, or on weekends.

This study has several limitations. First, 0–17 years was set as the younger age group, but it may not be appropriate to have this wide of an age range because of the large physiological differences between the ages. Second, in our registry, there were only 3 deaths in 0–17 years, although statistically significant, caution is needed in clinical confirmation. Third, in our study, we defined the nighttime from 6 pm to 9 am, which is study hospital's timetable and may vary depending on the country and region. Fourth, depending on the medical condition, TBI severity, and type of brain hemorrhage, the required treatment is different; therefore, the effect sizes of the off-hour effect may be different, but this was not considered. A subgroup analysis of SDH generally requiring surgical intervention has been performed, but it is still insufficient. Fifth, in addition to our study outcomes, the length of stay at ED and hospital may be a good outcome variable that shows the congestion of the ED or a smooth treatment process, however it was not included in the registry of our study. Finally, the study design was not a randomized controlled trial. Thus, there may be significant potential biases that were not controlled.

## Conclusions

In young patients under 18 years old with TBI with intracranial hemorrhage, admission during off-hours was associated with higher mortality at hospital discharge compared to that in those with admission during working hours.

## Author Contributions

**Data curation:** Hyun Ho Ryu.

**Formal analysis:** Hyun Ho Ryu.

**Funding acquisition:** Eujene Jung.

**Investigation:** Eujene Jung.

**Methodology:** Eujene Jung.

**Project administration:** Eujene Jung, Hyun Ho Ryu.

**Resources:** Eujene Jung, Hyun Ho Ryu.

**Software:** Eujene Jung.

**Supervision:** Eujene Jung.

**Writing – original draft:** Eujene Jung.

**Writing – review & editing:** Eujene Jung.

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
