## [Decision Letter · Decision Letter 0]

23 Aug 2022

PONE-D-22-13485The Off-Hour Effect on Mortality in Traumatic Brain Injury According to Age Group

PLOS ONE

Dear Dr. ryu,

Thank you for submitting your manuscript to PLOS ONE. After careful consideration, we feel that it has merit but does not fully meet PLOS ONE’s publication criteria as it currently stands. Therefore, we invite you to submit a revised version of the manuscript that addresses the points raised during the review process.

Dear Eujene, Jung and Hyun Ho, Ryu

Your manuscript “The Off-Hour Effect on Mortality in Traumatic Brain Injury According to Age Group” has now been assessed by our reviewers.

You have applied a simple analysis to the traumatic brain injury patients, and even though your findings may not be revolutionary – no insult intended – it is still of general interest.

The manuscript is therefore still considered for publication, although not in its current form. The two reviewers have raised some valid points, that may improve the manuscript and allow a revised version to be published. Please address all points, as well as the following:

Please ensure your names are spelled correctly throughout the manuscript (Ro is mentioned in Author Contributions, which I hope is a typo)Funding Acknowledgement: Please indicate whether the grant had an influence or otherwise restricted the research.Could you provide I little more information on EMT level-1 and level-2 differences in competencies?Please submit your revised manuscript by Oct 07 2022 11:59PM. If you will need more time than this to complete your revisions, please reply to this message or contact the journal office at plosone@plos.org. Please include the following items when submitting your revised manuscript:A rebuttal letter that responds to each point raised by the academic editor and reviewer(s). You should upload this letter as a separate file labeled 'Response to Reviewers'.A marked-up copy of your manuscript that highlights changes made to the original version. You should upload this as a separate file labeled 'Revised Manuscript with Track Changes'.An unmarked version of your revised paper without tracked changes. You should upload this as a separate file labeled 'Manuscript'.If applicable, we recommend that you deposit your laboratory protocols in protocols.io to enhance the reproducibility of your results. Protocols.io assigns your protocol its own identifier (DOI) so that it can be cited independently in the future. For instructions see: https://journals.plos.org/plosone/s/submission-guidelines#loc-laboratory-protocols. Additionally, PLOS ONE offers an option for publishing peer-reviewed Lab Protocol articles, which describe protocols hosted on protocols.io. Read more information on sharing protocols at https://plos.org/protocols?utm_medium=editorial-email&utm_source=authorletters&utm_campaign=protocols.

We look forward to receiving your revised manuscript.

Kind regards,

Tim Alex Lindskou

Academic Editor

PLOS ONE

Journal Requirements:

"This study was supported by a grant (BCRI-20022) of Chonnam National University Hospital Biomedical Research Institute"

"The funders had no role in study design, data collection and analysis, decision to publish, or preparation of the manuscript"

Reviewers' comments:

Reviewer's Responses to Questions

**Comments to the Author**

1. Is the manuscript technically sound, and do the data support the conclusions?

Reviewer #1: Yes

Reviewer #2: Partly

2. Has the statistical analysis been performed appropriately and rigorously? 

Reviewer #1: Yes

Reviewer #2: I Don't Know

3. Have the authors made all data underlying the findings in their manuscript fully available?

Reviewer #1: No

Reviewer #2: No

4. Is the manuscript presented in an intelligible fashion and written in standard English?

Reviewer #1: Yes

Reviewer #2: Yes

5. Review Comments to the Author

Reviewer #1: In a single center population of TBI patients between 2017-2020, the authors found no significant difference in mortality in patients visiting ED during off hours. The effect measure of ED admission during off hours on mortality was significant in patients aged 0-17 years compared to other age groups. Additionally, 85% of TBI patients presented during off hours and there were no differences in the proportion of surgical intervention between on- and off- hours.

Abstract:

- Would not state TBI is a surgical emergency, as 80% are mild and even moderate to severe may not always require surgery.

- Please include N. Please include how TBI was defined at the study institution. Please include more relevant data in the results section.

- I'm not sure the conclusion supports that any differences are currently present between daytime and nighttime emergency care.

Introduction:

- Would not state TBI is a surgical emergency, similar to above.

Methods:

- Authors need to define what constitutes "TBI" at their site. This alone could be a limitation depending on how it is defined.

- Per the authors, "The main outcome measure was mortality at hospital discharge. The secondary outcome was surgical intervention within 6 hours of ED admission. The main exposure variable was arrival time at ED admission (off-hour vs. working-hour). We defined daytime as 09:00 AM to 17:59 PM and nighttime as 18:00 PM to 08:59 AM; weekdays as Monday to Friday; and weekends as Saturday, Sunday, and national holidays, based on a previous study." As such, there should be a limitation discussed as 6 pm and 9 am are hardly accepted as "nighttime" at most institutions.

- Unclear why the authors controlled for so few variables while using a multivariable regression despite evaluating much larger initial cohorts. We know variables such as psychiatric history, history of prior TBIs, etc, can lead to poorer outcome.

- The Pediatric cohort is usually a completely separate population in characteristics and (in part) response to TBI, however were included as part of the main analysis in this study.

Limitations:

- Need to be augmented per comments above.

Discussion:

- The statement "EDs should, thus, be designed such that patients can expect the same quality of emergency care regardless of admission time." should be tempered/revised, as there is no recent evidence that ER care changes based on the hour.

- How can the results of this study change and/or improve practice?

Reviewer #2: 1. term seniors is. not universal (my pdf page 11)

2. weekdays and holidays are combined...are there Monday holidays like the US ?

3. am I correct these patients have isolated head injury. if they have a wrist fracture that only required splint, or a isolated ribfracture, were they excluded ?

4. is time from ED presentation to OR different in the weekday vs weekend group. are neurosurgeons available in the hospital during the day and night ? do they need to travel during these times from another office or home. I assume the OR staff/OR are 24/7/365.

5. maybe the important variable

6. table 1 ..what does intentionality mean

7. the p value ...what comparisons do they represent ...so age ...is it the 3 categories 0-17, 18-64 and ≥65 vs day and night? are these multiple comparisons still significant after a multiple comparison (Bonferroni) correction? same is true for all these other 3x2 , 5x2 ,4x2 comparisons

8. seems like there is overlaps in table 1 and 2....there is day vs night, weekday vs weekend, then in table 2 its really the same but now defined as off hours vs working hours. Why have the two and not just simply with on hours and off hours ?

9. it looks like these multiple comparisons also exist in other tables

so for example table 4 , mortality at hospital dc is p < .05. for a 3x2 table , 6 comparisons the level of significance is not .05 , but actually .05 /6 , significance adjusted 0.01250.

so your report of significance if not corrected at < .05 actually is. not significant between .01250 and .05.

6. PLOS authors have the option to publish the peer review history of their article (what does this mean?). If published, this will include your full peer review and any attached files.

Reviewer #1: No

Reviewer #2: **Yes: **Pierre Borczuk

---

## [Author Response · Author response to Decision Letter 0]

28 Sep 2022

PONE-D-22-13485

The Off-Hour Effect on Mortality in Traumatic Brain Injury According to Age Group

Journal: PLOS ONE

Editor comments

(1) Please ensure your names are spelled correctly throughout the manuscript (Ro is mentioned in Author Contributions, which I hope is a typo)

ANSWER: Thank you for review. We revised ‘Author Contributions’.

(2) Funding Acknowledgement: Please indicate whether the grant had an influence or otherwise restricted the research.

ANSWER: Thank you for review. We added funding statement within our cover letter.

(3) Could you provide I little more information on EMT level-1 and level-2 differences in competencies?

ANSWER: Thank you for review. We added some explanation the difference between level-1 and level-2 EMTs.

(REVISION: METHODS-Study design, setting, and population) The level-1 EMTs can be performed with high grade procedures including advanced airway and intravenous line access compared to level-2 EMTs

"The funders had no role in study design, data collection and analysis, decision to publish, or preparation of the manuscript"

ANSWER: Thank you for review. We remove funding-related text from the manuscript, update funding statement section, and added funding statement within our cover letter.

 

Reviewers' comments:

Reviewer #1: In a single center population of TBI patients between 2017-2020, the authors found no significant difference in mortality in patients visiting ED during off hours. The effect measure of ED admission during off hours on mortality was significant in patients aged 0-17 years compared to other age groups. Additionally, 85% of TBI patients presented during off hours and there were no differences in the proportion of surgical intervention between on- and off- hours.

Abstract:

- Would not state TBI is a surgical emergency, as 80% are mild and even moderate to severe may not always require surgery.

Answer: Thank you for the review. We edited the sentence according to your advice.

<Revision: Abstract-Background>

Traumatic brain injury (TBI) is a time-sensitive and life-threatening medical condition. 

- Please include N. Please include how TBI was defined at the study institution. Please include more relevant data in the results section.

Answer: Thank you for the review. We added the sentence according to your advice.

<Revision: Abstract-Results> 

: A total of 2086 patients with TBI with intracranial injury who were transported by EMS were enrolled in our registry.

: In the interaction analysis, the effect measure of ED admission during off-hours on mortality was significant among younger people (0—17 years: 1.17 [1.04—1.30]), compared to that in other age groups (18—64 years: 1.04 [0.46—2.37] and 65—100 years (0.97 [0.41—2.27])).

- I'm not sure the conclusion supports that any differences are currently present between daytime and nighttime emergency care.

Answer: Thank you for the review. Although a causal relationship has not been proven that there is a difference between emergency care and mortality when young TBI patients visit off-hours, based on previous studies showing that off-hour visit increased mortality and decreased surgical intervention, our study tried to prove that difference in mortality among off-hor visit differs according to age.

Introduction:

- Would not state TBI is a surgical emergency, similar to above.

Answer: Thank you for the review. We edited the sentence according to your advice.

<Revision: Introduction>

TBI is a time-sensitive and life-threatening medical condition

Methods:

- Authors need to define what constitutes "TBI" at their site. This alone could be a limitation depending on how it is defined.

Answer: Thank you for the review.

<Revision: Methods-Study desing, setting, and population>

The inclusion criteria are patients with TBI over 18 years of age who visited trauma center using emergency medical services (EMS) within 72 hours of injury and had intracranial hemorrhage and/or diffuse axonal confirmed by a radiological examination.

- Per the authors, "The main outcome measure was mortality at hospital discharge. The secondary outcome was surgical intervention within 6 hours of ED admission. The main exposure variable was arrival time at ED admission (off-hour vs. working-hour). We defined daytime as 09:00 AM to 17:59 PM and nighttime as 18:00 PM to 08:59 AM; weekdays as Monday to Friday; and weekends as Saturday, Sunday, and national holidays, based on a previous study." As such, there should be a limitation discussed as 6 pm and 9 am are hardly accepted as "nighttime" at most institutions.

Answer: Thank you for the review. We discussed in Limitation section according to your advice.

<Revision: Discussion>

The inclusion criteria are patients with TBI over 18 years of age who visited trauma center using emergency medical services (EMS) within 72 hours of injury and had intracranial hemorrhage and/or diffuse axonal confirmed by a radiological examination.

- Unclear why the authors controlled for so few variables while using a multivariable regression despite evaluating much larger initial cohorts. We know variables such as psychiatric history, history of prior TBIs, etc, can lead to poorer outcome.

Answer: Thank you for the review. We clearly defined the exclusion criteria for the study population according to your advice.

<Revision: Methods-Study desing, setting, and population>

Patients with neurological disorders, psychiatric disorders, and terminal cancer, pregnant women, patients with penetrating brain injury, and patients transferred to the participating hospital ED after surgery at another hospital were also excluded from the study.

- The Pediatric cohort is usually a completely separate population in characteristics and (in part) response to TBI, however were included as part of the main analysis in this study.

Answer: Thank you for the review. It is very important to point out that the incidence and progression patterns of pediatric TBI patients are somewhat different from those of adults. However, in our study, rather that looking at the difference in clinical outcomes of TBI between each age group, we look at the difference in clinical outcomes according to the time of ED visit in each age group. 

Limitations:

- Need to be augmented per comments above.

Answer: Thank you for the review. We added some sentences about limitations of our study according to your advice.

Discussion:

- The statement "EDs should, thus, be designed such that patients can expect the same quality of emergency care regardless of admission time." should be tempered/revised, as there is no recent evidence that ER care changes based on the hour.

Answer: Thank you for the review. We revised sentence according to your advice.

<Revision: Discussion>

it is necessary to pay more attention to that there is no difference in the quality of treatment regardless of the day and time of admission.

- How can the results of this study change and/or improve practice?

Answer: Thank you for the review. As described in the Discussion section of our study, the results of our study do not causally explain the decline in a quality of care for pediatric TBI patients at off-hours, but, it is theoretical basis for the need for pay more attention to that there is no difference in the quality of treatment regardless of the day and time of admission for the TBI patients, especially pediatric patients.

 

Reviewer #2: 

1. term seniors is. not universal (my pdf page 11)

Answer: Thank you for the review. We corrects the word according to your advice

<Revision: Introduction>

more experienced doctor

2. weekdays and holidays are combined...are there Monday holidays like the US ?

Answer: Thank you for the review. As you pointed out, the definition of weekday is clarified.

<Revision: Methods-Main outcome and variables>

weekdays as Monday to Friday except national holidays

3. am I correct these patients have isolated head injury. if they have a wrist fracture that only required splint, or a isolated ribfracture, were they excluded ?

Answer: Thank you for the review. As discussed in the Method section, patients who had TBI accompanied by injuried at other sites were excluded.

<Revision: Methods-Study design, setting, and population>

Patients who had TBI accompanied by injuries at other sites, those with no information of hospital admission, those who underwent surgical intervention, and those who died were excluded.

4. is time from ED presentation to OR different in the weekday vs weekend group. are neurosurgeons available in the hospital during the day and night ? do they need to travel during these times from another office or home. I assume the OR staff/OR are 24/7/365.

Answer: Thank you for the review. In principle, the level-1 trauma center has specialists such as neurosurgeon on standby 24/7/365, and personnel for surgical intervention are also on standby at the hospital. However, although not investigated in our study, we assumed that the number of personnel for surgical intervention and the activation process will not be smoother in off-hours. 

5. maybe the important variable

Answer: Thank you for the review. But we did not understand the meaning of your point. If possible, please explain again. 

6. table 1 ..what does intentionality mean

Answer: Thank you for the review. We added sentence about intentionality.

<Revision: Methods-Main outcome and variables>

intentionality (including self inflicted and interpersonal acts of violence intended to cause injury)

7. the p value ...what comparisons do they represent ...so age ...is it the 3 categories 0-17, 18-64 and ≥65 vs day and night? are these multiple comparisons still significant after a multiple comparison (Bonferroni) correction? same is true for all these other 3x2 , 5x2 ,4x2 comparisons

Answer: Thank you for the review. The p-values in Table 1 and 2 show the significance of the difference between each group. The p-values of the interaction analysis showe the significance of the odds ratios of the off-hour for each age when the working hour is used as a reference in each study outcomes. 

8. seems like there is overlaps in table 1 and 2....there is day vs night, weekday vs weekend, then in table 2 its really the same but now defined as off hours vs working hours. Why have the two and not just simply with on hours and off hours ?

Answer: Thank you for the review. Table 1 shows the characteristics of patients according to time of day and day of week, and Table 2 presents the differences in patient characteristics by dividing them into off-hour and working-hour. These definitions are clearly written in Method section.

<Revision: Methods-Main outcome and variables>

We defined off-hour when it was nighttime or weekends, and defined other times as working-hour.

9. it looks like these multiple comparisons also exist in other tables

so for example table 4 , mortality at hospital dc is p < .05. for a 3x2 table , 6 comparisons the level of significance is not .05 , but actually .05 /6 , significance adjusted 0.01250.

so your report of significance if not corrected at < .05 actually is. not significant between .01250 and .05.

Answer: Thank you for the review. We revised Table 4 according to your advice.

(Revision: Table 4)

Whole TBI patients Arrival time Mortality at hospital discharge Surgical intervention

Age 

 0-17 

 Working-hours reference reference

 Off-hours 1.17 (1.04-1.30) 1.52 (0.58-3.95)

 18-64 

 Working-hours reference reference

 Off-hours 1.04 (0.46-2.37) 1.04 (0.70-1.55)

 65- 

 Working-hours reference reference

 Off-hours 0.97 (0.41-2.27) 0.87 (0.56-1.35)

Subdural hemorrhage patients 

Age 

 0-17 

 Working-hours reference reference

 Off-hours 1.14 (0.97-1.33) 1.47 (0.57-3.81)

 18-64 

 Working-hours reference reference

 Off-hours 0.96 (0.43-2.17) 0.88 (0.60-1.30)

 65- 

 Working-hours reference reference

 Off-hours 1.09 (0.47-2.51) 0.87 (0.56-1.34)

---

## [Decision Letter · Decision Letter 1]

31 Oct 2022

PONE-D-22-13485R1The Off-Hour Effect on Mortality in Traumatic Brain Injury According to Age GroupPLOS ONE

Dear Dr. ryu,

Thank you for submitting your manuscript to PLOS ONE. After careful consideration, we feel that it has merit but does not fully meet PLOS ONE’s publication criteria as it currently stands. Therefore, we invite you to submit a revised version of the manuscript that addresses the points raised during the review process.

We look forward to receiving your revised manuscript.

Kind regards,

Tim Alex Lindskou

Academic Editor

PLOS ONE

Journal Requirements:

Reviewers' comments:

Reviewer's Responses to Questions

**Comments to the Author**

1. If the authors have adequately addressed your comments raised in a previous round of review and you feel that this manuscript is now acceptable for publication, you may indicate that here to bypass the “Comments to the Author” section, enter your conflict of interest statement in the “Confidential to Editor” section, and submit your "Accept" recommendation.

Reviewer #1: (No Response)

Reviewer #3: All comments have been addressed

2. Is the manuscript technically sound, and do the data support the conclusions?

Reviewer #1: No

Reviewer #3: Yes

3. Has the statistical analysis been performed appropriately and rigorously? 

Reviewer #1: No

Reviewer #3: Yes

4. Have the authors made all data underlying the findings in their manuscript fully available?

Reviewer #1: No

Reviewer #3: Yes

5. Is the manuscript presented in an intelligible fashion and written in standard English?

Reviewer #1: Yes

Reviewer #3: Yes

6. Review Comments to the Author

Reviewer #1: I appreciate the authors' time investment to address the Reviewer comments. However, a number of concerns remain:

1) The authors state in the Methods that TBI patients over the age of 18 were included, however the paper reports on patients aged 0-17 years.

2) I don't understand the rationale of doing a subgroup analysis in patients only with subdural hemorrhage. What about other types of intracranial injury?

3) I do not understand the rationale of having multiple significant differences in predictors on univariate analysis, and only including age and arrival time in multivariable analysis.

4) The study found that patients aged 0-17y had increased odds of mortality (Table 4) compared to other age groups when arriving during off-hours, yet in the conclusion does not provide a tangible explanation or actionable next step.

5) Off hour presentation with TBI, as the authors show, is associated with dangerous mechanisms (assaults, alcohol use) which require in depth discussions, especially relating to care received. While more 18-64yo patients presented during off hours, 0-17yo's had increased mortality. Why?

6) There were more SDH's during off hours than working hours. What about other forms of intracranial hemorrhage? This must be analyzed, controlled for and discussed thoroughly.

7) Are there differences in hospital lengths of stay between the two groups? This requires an adequate multivariable analysis.

Minor comments:

1) "The inclusion criteria are... and had intracranial hemorrhage and/or diffuse axonal [? injury]..."

Reviewer #3: Dear Editor,

Thank you for asking me to comment on the Revision -1 of this submission, although as you know I was not invited for the original review. So , my comments are brief since I see that the questions made by original reviews are being answered properly.

The only part that I am concerned is that the authors Excluded the patients under 18 ( inclusion of 18 and older). However their data and results are bout children less than 18 / How is this possible?

{{{{ The inclusion criteria are patients with TBI over 18 years of age who visited trauma center using

emergency medical services (EMS) within 72 hours of injury and had intracranial hemorrhage and/or

diffuse axonal confirmed by a radiological examination.}}}}

Best,

Ali Seifi Seifi, MD, FACP, FNCS, FCCM

Associate Professor of Neurocritical Care

University of Texas Health Science Center at San Antonio

7. PLOS authors have the option to publish the peer review history of their article (what does this mean?). If published, this will include your full peer review and any attached files.

Reviewer #1: No

Reviewer #3: No

---

## [Author Response · Author response to Decision Letter 1]

6 Nov 2022

Author’s reply to reviewers' comments:

On behalf of authors, thank you for the very valuable comments by the reviewer on our paper. We have attempted to address every point commented on by the reviewer in the revised manuscript. While we believe that we have addressed all of the reviewer’s concerns, we would be more than pleased to write additional revisions if needed.

We highlighted all changes in red. Author’s answers or explanations are in blue.

Correspondent author

Hyun Ho Ryu, MD, PhD

 

PONE-D-22-13485

The Off-Hour Effect on Mortality in Traumatic Brain Injury According to Age Group

Journal: PLOS ONE

Reviewer #1: 

1) The authors state in the Methods that TBI patients over the age of 18 were included, however the paper reports on patients aged 0-17 years.

ANSWER: Thank you for the review. There was an error in the description of the study population in 

our study, so it was corrected.

(REVISION: Methods-Study design, setting and population): The inclusion criteria are patients with TBI who visited trauma center using emergency medical services (EMS) within 72 hours of injury and had intracranial hemorrhage and/or diffuse axonal injury confirmed by a radiological examination.

2) I don't understand the rationale of doing a subgroup analysis in patients only with subdural hemorrhage. What about other types of intracranial injury?

ANSWER: Thank you for the review. In previous studies, it has been reported that Traumatic SDH among TBI is more time sensitive and shows a decisive difference in the prognosis of patients depending on whether or not surgical treatment is performed [1-3]. Also, Wilberger et al. noted that SDH patients who underwent surgery within four hours of injury may have a lower mortality rate and greater functional survival rates [4]. So, we performed subgroup analysis for patients with SDH, who are likely to be disadvantageous in definitive treatment including surgical treatment. We added description of subgroup analysis in the discussion. 

(REVISION: Discussion): SDH, which is reported to have a decisive effect on the prognosis of surgical treatment to decrease intracranial pressure, was assumed to be relatively disadvantage in off-hour compared to other brain injury, and a subgroup analysis was performed only on SDH patients [1-4].

3) I do not understand the rationale of having multiple significant differences in predictors on univariate analysis, and only including age and arrival time in multivariable analysis.

ANSWER: Thank you for the review. We adjusted the potential confounders in multivariable logistic regression analysis. In model 1, we adjusted age and sex and in model 2, we adjusted model 1 variables and intentionality, mechanism of injury, and place of injury were additionally adjusted. Table 3 and manuscript were modified. 

(REVISION: Methods-Statistical analysis): Multivariable logistic regression analyses were conducted to estimate the effect size of off-hours on mortality compared to that of working hours. Adjusted odds ratios (AORs) with corresponding 95% confidence intervals (CIs) were calculated after adjusting potential confounders. The potential confounders were selected based on directed acyclic graph (DAG) models.

4) The study found that patients aged 0-17y had increased odds of mortality (Table 4) compared to other age groups when arriving during off-hours, yet in the conclusion does not provide a tangible explanation or actionable next step.

ANSWER: Thank you for the review. We additionally commented on outcomes and next steps for young patients according to your advice.

(REVISION: Discussion): In this study, the off-hour effect was significantly associated with mortality at hospital discharge in the young patients under 17 year-old of age with TBI. To reduce the TBI burden, especially in young patients, additional manpower and equipment input in off-hours should be considered according to the circumstances of each hospital for maintained the quality of treatment regardless of the day and time of hospital admission.

5) Off hour presentation with TBI, as the authors show, is associated with dangerous mechanisms (assaults, alcohol use) which require in depth discussions, especially relating to care received. While more 18-64yo patients presented during off hours, 0-17yo's had increased mortality. Why?

ANSWER: Thank you for the review. The 18-64 years of aged population is less likely to develop injury during working hours. In addition, as outside activity increased during off-hours, and alcohol intake also increased significantly as suggested in this study, more TBI may have occurred. 

For pediatric patients, aged 0-17, as mentioned in the text, it can be assumed that the lack of resources during off-hours, such as manpower, could have more seriously affected young patients. We additionally commented in the discussion section.

(REVISION: Discussion): In this study, although the number of patients aged 18-64 years was relatively increased due to the increase in alcohol drink and the increase in outdoor activities during off-hours, the off-hour effect was significantly associated with mortality at hospital discharge in the young patients under 17 year-old of age with TBI.

6) There were more SDH's during off hours than working hours. What about other forms of intracranial hemorrhage? This must be analyzed, controlled for and discussed thoroughly.

ANSWER: Thank you for the review. SDH in off-hours was higher proportion than in working hours, but it was not statistically significant difference, and other forms of brain hemorrhage were additionally analyzed in Table 1 and Table 2 according to your advice.

(REVISION: Discussion and Table 1,2): In our study, although there was no statistically significant difference in the types of brain hemorrhage of patients visiting off-hours and working-hours, subgroup analysis for patients with SDH was performed, because the SDH proportion of off-hours was higher than working hours.

7) Are there differences in hospital lengths of stay between the two groups? This requires an adequate multivariable analysis.

ANSWER: Thank you for the review. As you pointed out, a length of stay could be a good clinical outcome for our study, but unfortunately, it was not collected in our registry. Relevant comments are explained in discussion (limitation) section. 

(REVISION: Discussion): Fourth, in addition to our study outcomes, the length of stay at ED and hospital may be a good outcome variable that shows the congestion of the ED or a smooth treatment process, however it was not included in the registry of our study.

Minor comments:

1) "The inclusion criteria are... and had intracranial hemorrhage and/or diffuse axonal [? injury]..."

ANSWER: Thank you for the review. We added missing words according to your advice.

(REVISION: Methods-Study design, setting and population): The inclusion criteria are patients with TBI who visited trauma center using emergency medical services (EMS) within 72 hours of injury and had intracranial hemorrhage and/or diffuse axonal injury confirmed by a radiological examination. 

Reviewer #3: 

(1) The only part that I am concerned is that the authors Excluded the patients under 18 ( inclusion of 18 and older). However their data and results are bout children less than 18 / How is this possible?

ANSWER: Thank you for the review. There was an error in the description of the study population in 

our study, so it was corrected.

(REVISION: Methods-Study design, setting and population): The inclusion criteria are patients with TBI who visited trauma center using emergency medical services (EMS) within 72 hours of injury and had intracranial hemorrhage and/or diffuse axonal injury confirmed by a radiological examination.

(REFERENCES)

1. Karibe, H.; Hayashi, T.; Hirano, T.; Kameyama, M.; Nakagawa, A.; Tominaga, T. Surgical management of traumatic acute subdural hematoma in adults: a review. Neurologia medico-chirurgica 2014, cr. 2014-0204.

2. Bullock, M.; Chestnut, R.; Ghajar, J. Guidelines for the surgical management of traumatic brain injury: surgical management of acute subdural hematomas. Neurosurgery 2006, 58, S16-S24.

3. Mathew, P.; Oluoch-Olunya, D.; Condon, B.; Bullock, R. Acute subdural haematoma in the conscious patient: outcome with initial non-operative management. Acta neurochirurgica 1993, 121, 100-108.

4. WILBERGER, J.E.J.; HARRIS, M.; DIAMOND, D.L. Acute Subdural Hematoma: Morbidity and Mortality Related to Timing of Operative Intervention. Journal of Trauma and Acute Care Surgery 1990, 30, 733-736.

---

## [Decision Letter · Decision Letter 2]

14 Dec 2022

PONE-D-22-13485R2The Off-Hour Effect on Mortality in Traumatic Brain Injury According to Age GroupPLOS ONE

Dear Dr. ryu,

Thank you for submitting your manuscript to PLOS ONE. After careful consideration, we feel that it has merit but does not fully meet PLOS ONE’s publication criteria as it currently stands. Therefore, we invite you to submit a revised version of the manuscript that addresses the points raised during the review process.

I do apologize for the long process time and understand the importance of publication this year. Therefore, I am sending you the comments following only a single review.

Please briefly address the last reviewers three comments, and then I am certain we can aim for a quick publication.

We look forward to receiving your revised manuscript.

Kind regards,

Tim Alex Lindskou

Academic Editor

PLOS ONE

Journal Requirements:

Reviewers' comments:

Reviewer's Responses to Questions

**Comments to the Author**

1. If the authors have adequately addressed your comments raised in a previous round of review and you feel that this manuscript is now acceptable for publication, you may indicate that here to bypass the “Comments to the Author” section, enter your conflict of interest statement in the “Confidential to Editor” section, and submit your "Accept" recommendation.

Reviewer #4: (No Response)

2. Is the manuscript technically sound, and do the data support the conclusions?

Reviewer #4: Yes

3. Has the statistical analysis been performed appropriately and rigorously? 

Reviewer #4: Yes

4. Have the authors made all data underlying the findings in their manuscript fully available?

Reviewer #4: Yes

5. Is the manuscript presented in an intelligible fashion and written in standard English?

Reviewer #4: Yes

6. Review Comments to the Author

Reviewer #4: Retrospective single center review examining outcomes for TBI patients by time off day and day of week of presentation. The authors demonstrate that outcomes are similar for all patients regardless of when they present. There is a signal for patients 0-17 where mortality seems to be increased at off-hours time, although this is based on only 3 deaths - and may statistically significant but clinically less so.

In general, outcomes are dependent on anatomic injury, presenting physiology, and comorbidities. Do the authors have any data on comorbidities in their study population (especially the older patients).

For the 0-17 age group, the findings are of interest, but the authors need to better describe specifically who cares for this patient population during the day as well as during off hours. Do you use pediatric specialists (and especially pediatric neurosurgeons and / or neurointensivists) and are any specialists available 24/ 7 or only during the workdays?

Finally, the conclusions should include the equivalent outcomes for the vast majority of patients. This speaks well of the trauma center / system that you have set up and should not be minimized.

7. PLOS authors have the option to publish the peer review history of their article (what does this mean?). If published, this will include your full peer review and any attached files.

Reviewer #4: No

---

## [Author Response · Author response to Decision Letter 2]

4 Jan 2023

Author’s reply to reviewers' comments:

On behalf of authors, thank you for the very valuable comments by the reviewer on our paper. We have attempted to address every point commented on by the reviewer in the revised manuscript. While we believe that we have addressed all of the reviewer’s concerns, we would be more than pleased to write additional revisions if needed.

We highlighted all changes in red. Author’s answers or explanations are in blue.

Thank you for your positive response to my request for expedited publication.

Correspondent author

Hyun Ho Ryu, MD, PhD

 

PONE-D-22-13485

The Off-Hour Effect on Mortality in Traumatic Brain Injury According to Age Group

Journal: PLOS ONE

Reviewer #4: 

(1) Retrospective single center review examining outcomes for TBI patients by time off day and day of week of presentation. The authors demonstrate that outcomes are similar for all patients regardless of when they present. There is a signal for patients 0-17 where mortality seems to be increased at off-hours time, although this is based on only 3 deaths - and may statistically significant but clinically less so.

ANSWER: Thank you for the review. We agree with the fact that the result of our study that the mortality at 0—17-year-old increases in statistically significant, but clinically has limitations, and we explained the limitation section.

(REVISION: Discussion): Second, in our registry, there were only 3 deaths in 0—17 years, although statistically significant, caution is needed in clinical confirmation.

(2) In general, outcomes are dependent on anatomic injury, presenting physiology, and comorbidities. Do the authors have any data on comorbidities in their study population (especially the older patients).

ANSWER: Thank you for the review. We added variables of comorbidities including hypertension and diabetes mellitus in multivariable logistic regression analysis according to your advice.

(REVISION: Table 1,2,3)

(3) For the 0-17 age group, the findings are of interest, but the authors need to better describe specifically who cares for this patient population during the day as well as during off hours. Do you use pediatric specialists (and especially pediatric neurosurgeons and / or neurointensivists) and are any specialists available 24/ 7 or only during the workdays?

ANSWER: Thank you for the review. Our trauma center does not have a specialist in charge of only pediatrics, and cares by department regardless of age. I add these to the method section.

(REVIOSN: Methods-Study design, setting, and population): These protocols are the same regardless of age, and there is no medical staff dedicated to a specific age group.

(4) Finally, the conclusions should include the equivalent outcomes for the vast majority of patients. This speaks well of the trauma center / system that you have set up and should not be minimized.

ANSWER: Thank you for the review. As you commented, the conclusions of our study are limited to generalization because they are the results of the EMS and ED environments in Korea. We deleted this sentence (‘EDs should, thus, be designed such that patients can expect the same quality of emergency care regardless of admission time.’) according to your advice.

---

## [Editor Report · Decision Letter 3]

28 Feb 2023

The Off-Hour Effect on Mortality in Traumatic Brain Injury According to Age Group

PONE-D-22-13485R3

Dear Dr. ryu,

We’re pleased to inform you that your manuscript has been judged scientifically suitable for publication and will be formally accepted for publication once it meets all outstanding technical requirements.

Kind regards,

Tim Alex Lindskou

Academic Editor

PLOS ONE

Additional Editor Comments (optional):

Thank you for the revised manuscript.

It appears there have been some latency from your submission of the latest revision, to I received it. This is completely out of my hands, but still unfortunate.

Well done on your article and for your patience.
---

## [Editor Report · Acceptance letter]

9 Mar 2023

PONE-D-22-13485R3 

The Off-Hour Effect on Mortality in Traumatic Brain Injury According to Age Group 

Dear Dr. Ryu:

I'm pleased to inform you that your manuscript has been deemed suitable for publication in PLOS ONE. Congratulations! Your manuscript is now with our production department. 

Kind regards, 

on behalf of

Dr. Tim Alex Lindskou 

Academic Editor

PLOS ONE